



# A simple method for retrieval of dust aerosol optical depth with polarized reflectance over oceans

Wenbo Sun[1,3,*], Yongxiang Hu[2], Rosemary R. Baize[2], Gorden Videen[3,4], Sungsoo S. Kim[3], Young-Jun Choi[5], Kyungin Kang[6], Chae Kyung Sim[3], Minsup Jeong[5], Ali Omar2, Snorre A. Stamnes[2], and Evgenij Zubko[7]

[1]Science Systems and Applications Inc, Hampton, VA 23666, USA
[2]NASA Langley Research Center, Hampton, VA 23681, USA
[3]Kyung Hee University, Yongin-shi, Kyungki-do 17104, Korea
[4]US Army Research Lab, Adelphi Maryland 20783, USA
[5]Korea Astronomy and Space Science Institute, Yuseong-gu, Daejeon 34055, Korea
[6]Korea Advanced Institute of Science and Technology, Daejeon 34141, Korea
[7]Far Eastern Federal University, Vladivostok 690950, Russia
[*]1 Enterprise Parkway, Hampton, VA 23681, USA
wenbo.sun-1@nasa.gov

**Abstract.** Our previous study shows that the angle of linear polarization (AOLP) of solar radiation that is scattered from clouds at near-backscatter angles can be used to detect super-thin cirrus clouds over oceans. Such clouds are too thin to be sensed using any current passive satellite instruments that only measure light's total intensity, because of the uncertainty in surface reflection. In this report, we show that with a method similar to the super-thin clouds detection algorithm, dust aerosols may also be detected and differentiated from clouds. We also show that the degree of polarization of reflected light can be used for retrieving the optical depth of dust aerosols in the neighborhood of the backscatter angle, regardless of the reflecting surface conditions. This is a simple and robust algorithm, which could be used to survey dust aerosols over midlatitude and tropical oceans.

**Key words:** Polarized reflectance; degree of polarization; dust aerosol; retrieval; remote sensing.

A NASA-Korea CubeSat mission is currently under preparation by NASA Langley Research Center, the Korea Astronomy and Space Science Institute (KASI), and Kyung Hee University of Korea. We plan to use polarimeters on two CubeSats to detect the super-thin clouds over global oceans and dust aerosols over oceans and land around the Korean peninsula. The polarimeters will be developed by KASI, that are modified versions of the Polarimetric Camera (PolCam) developed by KASI for the Korea Pathfinder Lunar Orbiter (KPLO). This planned polarimeter-on-CubeSat mission will measure the polarization features of scattered light from clouds and aerosols to identify the super-thin clouds and dust aerosols over oceans and retrieve their optical depth.

Our previous works (Sun et al. 2014; 2015) show that distinct features exist in the angle of linear polarization (AOLP) of solar radiation that is scattered from clouds at near-backscatter angles. At these angles the dominant electric field from clear-sky oceans is nearly parallel to the Earth surface. However, when clouds are present, this electric field can rotate significantly away from the parallel direction. Our modeling results suggest that this polarization feature can be used to detect super-thin cirrus clouds having an optical depth of only ~0.06 and super-thin liquid water clouds having an optical depth of only ~0.01. Such clouds are too thin to be sensed using any current passive satellite instruments that only measure light's total intensity, because of the uncertainty in surface reflection.



Similar to super-thin clouds, aerosols such as dust particles also affect surface remote sensing
and global climate significantly. The optically thin aerosols are also very difficult to detect even
over a dark surface condition such as oceans. In optical remote sensing of aerosols, how to
distinguish between surface and atmospheric contributions to the TOA reflectance keeps on a
problem. Because of aerosols' small optical thickness, the uncertain effect of ocean surface
reflection to the light cannot be well quantified when using total reflectance as a measurement to
the aerosols, even with multiple angles and wavelengths in measurements. Several methods are
proposed to separate the atmospheric and the surface contributions (e.g., Kaufman et al. 1997),
but no ideal method is reported to date. The use of the degree of polarization of the radiance is
thought to have great potential for aerosol retrieval (Herman et al., 1997). However, no robust
method for remote sensing of aerosols based on polarized radiation measurement is reported to
date. Based on our modeling results, we will propose a novel method of using passive
polarimetric instruments to detect dust aerosols over oceans in this paper.
Unpolarized solar radiation can be polarized by surface reflections as well as by scattering from
atmospheric molecules and particles. When sunlight propagates through a clear atmosphere and
is scattered back toward the Sun, the resulting signal is nearly unpolarized when the solar zenith
angle (SZA) is less than ~40° (Sun and Lukashin, 2013). By considering a longer solar
wavelength, such as 670 nm, the contribution of molecular scattering is small. Unlike total
radiance ($I$), the degree of polarization (DOP) and angle of linear polarization (AOLP) of the
reflected sunlight are insensitive to surface roughness and absorption by atmospheric water vapor
and other gases (Sun and Lukashin, 2013). This insensitivity makes the polarization
measurement robust for different environmental conditions, even when the detected components
are within the lower layers of the atmosphere. For example, super-thin water clouds close to the
surface of the Earth that cannot be detected using 1.38µm radiance can be identified by the
polarization properties of light backscattered from them. Method for using AOLP feature to
detect super-thin clouds is reported in Sun et al. (2014). The method for using polarized
reflectance to retrieve the optical depth of super-thin clouds is reported in Sun et al. (2015). Sun
et al. (2015) reports that the optical depth of super-thin clouds can be retrieved at near-
backscatter angles without the effect of background reflection.
Our studies show that the polarization of backscattered light can also be applied to aerosol
remote sensing. Figure 1 shows the modeled DOP of reflected sunlight at 670 nm from dust
aerosols over oceans. Also shown in this figure are results from 12 days of PARASOL level-1
reflectance and level-2 ocean aerosol and clouds data (Deschamps et al.  1994; Buriez et al.
1997; Tanre et al. 2011) across May to August of 2006. In the modeling, we assume the dust
particles are nonspherical debris aggregates with a refractive index of 1.4 + 0.01i (Zubko et al.
2006; 2009; 2013). The aerosols are within a 1-km layer over ocean surface. The aerosol size
distribution and single-scattering property calculation follow those reported in Sun et al. (2013).
The PARASOL measurements obtained over Atlantic Ocean area (0°N -35°N and 0°W -60°W)
are used to capture Sahara dust over oceans. Only those data with an AOD = 0.2 − 0.4 from the
PARASOL OC2 dataset are used for comparison with the modeled results. We can see that the
modeled DOP of reflected light is a strong function of dust AOD. At the near-backscatter
viewing angles, DOP of reflected light monotonically increases with the AOD. This means that
when using a polarimeter at these observation angles, AOD can be retrieved from the DOP of the
backscattered light. The PARASOL data well prove the modeled results at the near-backscatter
angles. However, significant difference is found at other viewing angles, with unknown reasons.





Figure 2 shows the AOLP of the reflected light from the ADRTM (left panel, AOD = 0.3) and
the PARASOL (right panel, AOD = 0.2 − 0.4). We can see that at near-backscatter angles,
AOLPs from the model and satellite data are significantly different. The PARASOL results have
a glory pattern at near-backscatter angles that indicates transparent cloud particles such as water
droplets or ice crystals (Sun et al. 2014; 2015). This means the PARASOL OC2 aerosol product
has clouds contamination. Thus, the aerosol properties in the PARASOL OC2 may not be very
reliable for this case.

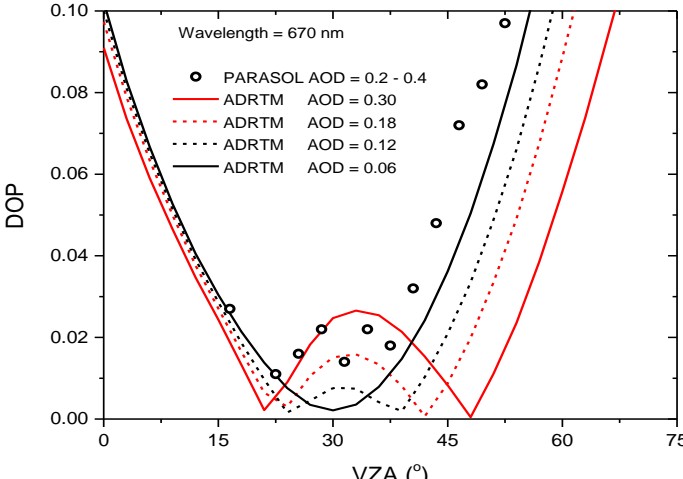

Figure 1. At near-backscatter angles [For this case, it is at viewing zenith angle (VZA) = ~20-40º,
and relative azimuth angle (RAZ) = ~170-180º], the DOP of light from dust monotonically
increases with the AOD of dust aerosols. In this modeling, the adding-doubling radiative transfer
model (ADRTM) developed in Sun and Lukashin (2013) is used. The wavelength is 670 nm, the
solar zenith angle (SZA) is 30º, and wind speed over ocean surface is 7.5 m/s. Also shown in this
figure are the PARASOL measurements obtained over Atlantic Ocean area (0ºN -35ºN and 0ºW -
60ºW) that has Sahara dust. 12 days of PARASOL data in May-August, 2018 are used for this
study. The results are for a RAZ of 177º.

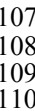

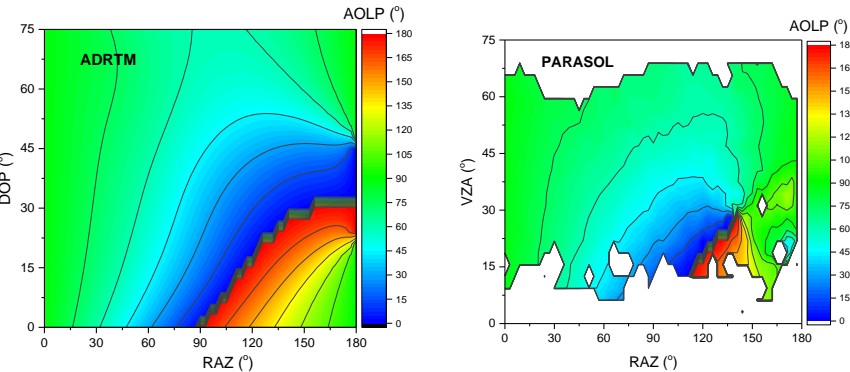




Figure 2. Same as in Fig. 1, but for the AOLP of the reflected light from the ADRTM (left panel,
AOD = 0.3) and the PARASOL (right panel, AOD = 0.2 – 0.4).

In summary, our modeling results show that the DOP of scattered sunlight can be used to detect
aerosols. Figure 2 also shows that the AOLP of scattered light from nonspherical dust particles
are very different from that of light scattered by clouds as reported in Sun et al. (2014; 2015).
This can be used to differentiate aerosols from clouds, regardless of the ocean surface conditions.
This is a simple and robust algorithm, which could be used to survey dust aerosols over
midlatitude and tropical oceans, as planned by the NASA-Korea CubeSat mission for the
detection of super-thin clouds/aerosols.

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
