# Peer review of "A simple method for retrieval of dust aerosol optical depth with polarized reflectance over oceans"

_Atmospheric Chemistry and Physics, 2019_

## Referee Comment (RC1) · Anonymous Referee #1 · 1 Nov 2019

Comments on the manuscript entitled "A simple method for retrieval of dust aerosol optical depth with polarized reflectance over oceans" by Wenbo Sun, et al.

General comments: The current passive satellite instruments can only measure the total intensity of solar radiation, which couldn't detect the optically super-thin cirrus clouds due to the uncertainty in surface reflection. This manuscript proposes a novel and robust algorithm of using passive polarimeter and can detect the super-thin clouds and dust aerosols. The optical depth of dust aerosols in the neighborhood of the backscatter angle can be also retrieved by using the degree of polarization of reflected light, regardless of the reflecting surface conditions. This novel method is expected to be

used in the planned NASA-Korea CubeSat mission for detecting the super-thin clouds and dust aerosol over midlatitude and tropical oceans. Overall, I think the idea and algorithm proposed by this manuscript are innovative and the English writing is fine, and I recommend this manuscript is appropriate for publishing after minor revision.

Minor comments: 1. Page 2, lines 81-83: "In the modeling, we assume the dust particles are nonspherical debris aggregates with a refractive index of 1.4 + 0.01i (Zubko et al. 2006; 2009; 2013)" âĞŠ Please explain briefly the reason of the refractive index (1.4 + 0.01i) of dust particles assumed in the model, whether the selection of different refractive index will affect the modeling results.

2. Page 2, lines 79-81: "Also shown in the figure are results from 12 days of PARASOL level-1 reflectance and level-2 ocean aerosol and clouds data (Deschamps et al. 1994; Buriez et al. 1997; Tanre et al. 2011) across May to August of 2006." Page 3, Figure 1, line 107: "12 days of PARASOL data in May-August, 2018 are used for this study." âĞŠ The date of PARASOL data used in this manuscript should be the same. Please check it.

---

## Referee Comment (RC2) · Anonymous Referee #2 · 5 Nov 2019

The paper "A simple method for retrieval of dust aerosol optical depth with polarized reflectance over oceans" is aimed at the development of the technique to retrieve dust aerosol optical depth using spaceborne observations. Unfortunately I can not recommend this paper for publication. Actually the authors do not describe their technique to solve the inverse problem in the paper. They also do not show the validation results. They a state that there is no robust method for remote sensing of aerosols based on polarized radiation measurements. This is not true ( please, see the papers by Dubovik, Hasekamp, Herman, etc.).

---

## Author Response (AR1)

**Point-by-point response to the reviews, a list of all relevant changes made in the**
**manuscript of "A simple method for retrieval of dust aerosol optical depth with polarized**
**reflectance over oceans" by Wenbo Sun et al.**
**Response to Referee 1**
*General comments: The current passive satellite instruments can only measure the total intensity*
*of solar radiation, which couldn't detect the optically super-thin cirrus clouds due to the*
*uncertainty in surface reflection. This manuscript proposes a novel and robust algorithm of*
*using passive polarimeter and can detect the super-thin clouds and dust aerosols. The optical*
*depth of dust aerosols in the neighborhood of the backscatter angle can be also retrieved by*
*using the degree of polarization of reflected light, regardless of the reflecting surface conditions.*
*This novel method is expected to be used in the planned NASA-Korea CubeSat mission for*
*detecting the super-thin clouds and dust aerosol over midlatitude and tropical oceans. Overall, I*
*think the idea and algorithm proposed by this manuscript are innovative and the English writing*
*is fine, and I recommend this manuscript is appropriate for publishing after minor revision.*
**The authors thank this reviewer for the helpful comments. The manuscript was revised**
**following the comments rigorously.**
*Minor comments: 1. Page 2, lines 81-83: "In the modeling, we assume the dust particles are*
*nonspherical debris aggregates with a refractive index of 1.4 + 0.01i (Zubko >et al. 2006; 2009;*
*2013)" Please explain briefly the reason of the refractive index (1.4 + 0.01i) of dust particles*
*assumed in the model, whether the selection of different refractive index will affect the modeling*
*results.*
**Since dust refractive index has uncertainty due to different components and moisture, we**
**chose this representative refractive index just for demonstration of the idea of the method,**
**this will affect the modeling result a little bit (especially when imaginary part is very big),**
**but not affect any conclusions in the paper.**
*2. Page 2, lines 79-81: "Also shown in the figure are results from 12 days of PARASOL level-1*
*reflectance and level-2 ocean aerosol and clouds data (Deschamps et al. 1994; Buriez et al.*
*1997; Tanre et al. 2011) across May to August of 2006." Page 3, Figure 1, line 107: "12 days of*
*PARASOL data in May-August, 2018 are used for this study." The date of PARASOL data used*
*in this manuscript should be the same. Please check it.*
**This is a typo, "2018" should be "2006". We corrected it.**
**Response to Referee 2**
*The paper "A simple method for retrieval of dust aerosol optical depth with polarized*
*reflectance over oceans" is aimed at the development of the technique to retrieve dust aerosol*
*optical depth using spaceborne observations. Unfortunately I can not recommend this paper for*
*publication. Actually the authors do not describe their technique to solve the inverse problem in*

*the paper. They also do not show the validation results. They a state that there is no robust method for remote sensing of aerosols based on polarized radiation measurements. This is not true ( please, see the papers by Dubovik, Hasekamp, Herman, etc.).*

**The authors thank this reviewer for the helpful comments and followed the reviewer's comments to correct the manuscript rigorously.**

*Actually the authors do not describe their technique to solve the inverse problem in the paper. They also do not show the validation results.*

**Our Fig. 1 has clearly showed that dust aerosol OD can be simply retrieved using the DOP in the neighborhood of backscatter angle. PARASOL data in the same figure validated the results. However, for a detailed algorithm, we are planning a full article after our instrument's data are obtained. This short letter has no intention to report full technical algorithm, but a simple idea.**

*They a state that there is no robust method for remote sensing of aerosols based on polarized radiation measurements. This is not true ( please, see the papers by Dubovik, Hasekamp, Herman, etc.).*

**To follow this comment, we changed our statement to "However, the retrieval method of remote sensing of aerosols based on polarized radiation measurement is still in progress (Dubovik et al. 2019)." And cited:**

**Dubovik, O., Li, Z., Mishchenko, M. I., Tanré, D., Karol, Y., Bojkov, B., Cairns, B., Diner, D. J., Espinosa, W. R., Goloub, P., Gu, X., Hasekamp, O., Hong, J., Hou, W., Knobelspiesse, K. D., Landgraf, J., Li, L., Litvinov, P., Liu, Y., Lopatin, A., Marbach, T., Maring, H., Martins, V., Meijer, Y., Milinevsky, G., Mukai, S., Parol, F., Qiao, Y., Remer, L., Rietjens, J., Sano, I., Stammes, P., Stamnes, S., Sun, X., Tabary, P., Travis, L. D., Waquet, F., Xu, F., Yan, C., and Yin, D.: Polarimetric remote sensing of atmospheric aerosols: instruments, methodologies, results, and perspectives, J. Quant. Spectrosc. Radiat. Transfer 224, 474-511, doi:10.1016/j.jqsrt.2018.11.024, 2019.**

**This review article reports the current status of polarization retrieval algorithms.**

**Other changes**

**We also changed the presentation VZA angle range of Fig. 1 to make the figure more symmetric. (No data are changed).**

**We inserted an Acknowledgment at end of the paper.**

[revised manuscript text omitted]

---

## Author Response (AR2)

**Co-Editor Decision: Publish subject to technical corrections** (28 Nov 2019) by Jianping Huang
Comments to the Author:
Considering the situation of manuscript and reviewers' comments, the manuscript appears more suitable for final publication as a Technical note.
Please adjust the manuscript category and revise its title to begin with "Technical note".

Response:

Thanks the editor for the work on this paper, the manuscript has been modified following the request.